# Around one third of current Arctic Ocean primary production sustained by rivers and coastal erosion

Jens Terhaar [1,2,3,4 ✉], Ronny Lauerwald [1,2,5], Pierre Regnier[2], Nicolas Gruber [6] & Laurent Bopp[7]

Net primary production (NPP) is the foundation of the oceans' ecosystems and the fisheries they support. In the Arctic Ocean, NPP is controlled by a complex interplay of light and nutrients supplied by upwelling as well as lateral inflows from adjacent oceans and land. But so far, the role of the input from land by rivers and coastal erosion has not been given much attention. Here, by upscaling observations from the six largest rivers and using measured coastal erosion rates, we construct a pan-Arctic, spatio-temporally resolved estimate of the land input of carbon and nutrients to the Arctic Ocean. Using an ocean-biogeochemical model, we estimate that this input fuels 28–51% of the current annual Arctic Ocean NPP. This strong enhancement of NPP is a consequence of efficient recycling of the land-derived nutrients on the vast Arctic shelves. Our results thus suggest that nutrient input from the land is a key process that will affect the future evolution of Arctic Ocean NPP.

[1] Laboratoire des Sciences du Climat et de l'Environnement, LSCE/IPSL, CEA-CNRS-UVSQ, Université Paris-Saclay, 91191 Gif-sur-Yvette, France. [2] Biogeochemistry and Earth System Modelling, Department of Geoscience, Environment and Society, Université Libre de Bruxelles, Bruxelles, Belgium. [3] Climate and Environmental Physics, Physics Institute, University of Bern, Bern, Switzerland. [4] Oeschger Center for Climate Change Research, University of Bern, Bern, Switzerland. [5] Université Paris-Saclay, INRAE, AgroParisTech, UMR ECOSYS, 78850 Thiverval-Grignon, France. [6] Environmental Physics, Institute of Biogeochemistry and Pollutant Dynamics, ETH Zurich, Zurich, Switzerland. [7] LMD/IPSL, Ecole Normale Supérieure/PSL University, CNRS, Ecole Polytechnique, Sorbonne Université, Paris, France. ✉email: jens.terhaar@climate.unibe.ch

Primary production in the Arctic Ocean by unicellular phytoplankton forms the basis of a unique ecosystem that supports a rich wildlife with some of Earth's most iconic top predators, such as polar bears or walrus[1]. The nutrients supporting this marine primary production, and especially the limiting nutrient nitrogen[2,3], are believed to stem largely from lateral input from the adjacent oceans and from upwelling and mixing from below[4]. Not well known is the role of the nutrients supplied to the Arctic Ocean from land via rivers and through the erosion of coastal soils. Understanding the role of these terrigenous nutrients on Arctic Ocean NPP is of paramount importance, especially as the Arctic Ocean and its catchment are one of the world's fastest-changing regions, mostly due to anthropogenic climate change[5].

It has been estimated that Arctic Ocean NPP increased by 57% between 1998 and 2018 due to warming, sea-ice reduction, and changes in ocean circulation[6,7]. Model simulations have suggested that this increase will continue in the 21st century[8] and has the potential to significantly enhance fisheries catch in the Arctic Ocean[9]. However, the models used for these projections differ strongly in their simulated present-day[10] and future Arctic Ocean NPP[8] due to the complex interplay of nutrient and light limitation of phytoplankton production. Differences between these models, as well as observed changes in NPP[6,7,11] have mainly been related to the specific physical conditions of the Arctic Ocean, such as sea-ice extent and ocean circulation. However, recent studies have suggested that terrigenous nutrient inputs from rivers and coastal erosion might be another key control of Arctic Ocean NPP[12–14], a process that has often been neglected in observational studies[6,7,11] and models[8,10].

Neglecting the role of terrigenous nutrients is particularly problematic in the Arctic Ocean, as their impact on marine NPP is presumably large compared to other ocean regions due to the Arctic Ocean's unique geographical setting. The Arctic Ocean is the only ocean that has a watershed area that is larger than its own area. It receives around 11% of global river discharge although it holds only 1% of the global ocean volume[15]. In addition, the Arctic coastline is eroding fast due to thawing permafrost, providing another important source of terrigenous nutrients[13,16].

Despite the potential importance of terrigenous nutrient inputs for Arctic Ocean NPP, the magnitude of these fluxes[12,16,17] and their net impact on Arctic Ocean NPP are not well known[13,18–20]. Measurements of the riverine nutrient fluxes were taken frequently at the six largest Arctic rivers (Mackenzie, Yukon, Kolyma, Lena, Ob, Yenisei)[12,21,22], and were periodically recorded in several smaller river systems[23–26], but few time series of observations were collected elsewhere. Thus, to estimate the total riverine fluxes of carbon and nutrients into the Arctic Ocean, the fluxes from the six largest rivers were extrapolated to the entire Arctic catchment[18,20,21,27]. Riverine nutrient inputs determined from previously published pan-Arctic organic carbon fluxes were used to estimate riverine-driven NPP to vary between 4[18] and 10%[19,20] of total Arctic NPP. In contrast to riverine nutrient fluxes, no estimates of nutrient fluxes from coastal erosion and of their impact on Arctic NPP exist yet, mainly because of the limited number of observations of nutrient content in the eroded soils[28,29].

Here we provide (1) a gridded estimate of the seasonally varying river input of dissolved nutrients and carbon, (2) a gridded estimate of the seasonally varying input of nutrients and carbon from coastal erosion and (3) a quantitative assessment of the importance of these two sources of nutrients for Arctic Ocean NPP. To do so, we derived a monthly, spatially resolved, observation-based forcing set for riverine dissolved terrigenous carbon and nutrient inputs to the coastline north of 60°N.

These riverine fluxes are based on observed monthly fluxes from the six largest Arctic rivers[12,21], which were extrapolated to unmonitored river basins using a spatially explicit prediction based on watershed characteristics (see Methods). For coastal erosion, total (particulate + dissolved) terrigenous carbon and nutrient inputs were calculated as the product of spatially resolved erosion rates[30] and estimates of total carbon and nutrient content in coastal soils[28–31]. The riverine and coastal erosion fluxes of nutrients and carbon were then used to force the global ocean-biogeochemical model NEMO-PISCES[32] at high-resolution (~14 km in the Arctic Ocean) over the period 1990–2010 (Baseline simulation). The version of NEMO-PISCES used here does not allow us to add organic matter fluxes with carbon-to-nitrogen (C:N) and carbon-to-phosphorus ratios that differ from the stoichiometric marine ratios in the model. Therefore, riverine and coastal erosion fluxes were added in their inorganic form assuming immediate remineralisation of all organic matter at the land-ocean interface (see Methods). We also do not consider the effect of terrigenous particulate matter emanating from the rivers or coastal erosion on turbidity of the waters, and hence the light availability for phytoplankton growth. We later assess the uncertainty and potential bias related to these assumptions and their effect on simulated Arctic Ocean NPP.

In addition to the baseline simulation, we also conducted a simulation without any terrigenous input (Referred to as NoTerr) north of 60°N and one where only the river input was considered (NoCoast). The differences between these three simulations permit us then to quantify the impact of the various forms of terrigenous input. Throughout the rest of the manuscript, we focus on nitrogen given its role as the limiting nutrient for NPP in the Arctic Ocean[2,3] (Supplementary Fig. 1).

With these simulations, we estimate that terrigenous nutrients fuel 28–51% of the current annual Arctic Ocean NPP. Our results thus suggest that nutrient input from the land is a key process for Arctic Ocean NPP and its future evolution.

## Results and discussion

**Nitrogen input from rivers.** We estimate that rivers currently deliver 1.0 (0.9–1.1) Tg N yr$^{-1}$ of dissolved nitrogen (inorganic 32% and organic 68%) to the Arctic Ocean, which is limited here by the Fram Strait, the Barents Sea Opening, the Bering Strait, and the Baffin Bay (Table 1; Supplementary Fig. 2). Sixty percent of the dissolved nitrogen that enters the Arctic Ocean by rivers is provided by only five rivers (Mackenzie, Pechora, Ob,

**Table 1 Annual carbon and nutrient fluxes from rivers and coastal erosion with one standard deviation provided in brackets (see methods).**

|  | This study | Previous estimates |
|---|---|---|
| **Riverine inputs** |  |  |
| Dissolved inorganic carbon | 50.6 (45.0–56.2) | 46.0 ± 7.0[21] |
| Dissolved organic carbon | 27.6 (24.6–30.6) | 25–30[12,27,78] |
| Dissolved inorganic nitrogen | 0.32 (0.28–0.36) | 0.32[12] |
| Dissolved organic nitrogen | 0.67 (0.60–0.74) | 0.63[12] |
| Dissolved (inorg. & org.) phosphorus | 0.06 (0.05–0.07) | 0.05[12] |
| Silicate | 8.8 (7.8–9.8) | 8.38[12] |
| **Coastal erosion inputs** |  |  |
| Total carbon | 15.4 (9.2–24.2) | 4.9–14.0[17] |
| Total nitrogen | 1.6 (1.0–2.5) | — |
| Total phosphorus | 0.27 (0.16–0.42) | — |

Units are in Tg yr$^{-1}$.

Yenisei, Lena) (Fig. 1), and with four of them located in the Russian Arctic, this region dominates the input. The seasonality of the dissolved nitrogen input is strong. During winter, riverine delivery keeps a baseflow of 0.03 Tg N mon$^{-1}$ (Fig. 2) before increasing during spring freshet to a peak in June (0.33 Tg N mon$^{-1}$) and then decreasing back to winter baseflow[12]. Rivers also deliver particulate organic nitrogen, amounting to around 0.7 Tg N yr$^{-1}$ [22]. However, 85–95% of this particulate nitrogen is believed to settle in the river delta[33]. Given the likely limited impact and the still high uncertainties associated to the biogeochemical cycling of particulate matter across the river-estuary continuum[22], we did not include any riverine input of particulate organic matter in our simulations, but later discuss this additional input and its potential role for Arctic Ocean NPP.

Our dissolved riverine nitrogen fluxes to the Arctic Ocean are in agreement with previous estimates that extrapolated the nutrient fluxes of the six largest Arctic rivers by applying the average yield to unmonitored watersheds[12]. The agreement is expected as both estimates rely on the same observed flux data and corroborates the use of average yield to upscale fluxes to the pan-Arctic scale. Note, however, that an average yield would probably not reproduce the spatial heterogeneity of nutrient input along the Arctic coastline as it is achieved in our estimate (Fig. 1).

**Nitrogen input from coastal erosion.** For coastal erosion, we estimate a delivery of 1.6 (1.0–2.5) Tg N yr$^{-1}$ of total nitrogen to the Arctic Ocean (Table 1). As opposed to rivers, this nitrogen input occurs almost entirely in particulate organic form[34]. The spatial distribution of nitrogen input from coastal erosion is more homogeneous, with highest rates along the Eurasian coastline (Fig. 1). Fluxes from coastal erosion peak later in the year (Fig. 2), i.e. they are strongest in August and September (0.45 Tg N mon$^{-1}$), when sea ice reaches its minimum and coastlines are exposed to wind and waves, and negligible in winter[35].

No previous pan-Arctic nitrogen fluxes from coastal erosion exist yet to compare our estimates against. However, we can compare our coastal erosion carbon fluxes, from which the nitrogen fluxes are derived via a fixed C:N ratio, to previous estimates[17]. These previous pan-Arctic carbon flux estimates from coastal erosion vary from 4.9 to 14.0 Tg C yr$^{-1}$ with the most recent study being at the high end[16]. The here presented estimate of 15.4 (9.2–24.2) Tg C yr$^{-1}$ agrees with the previous estimates within the uncertainties (Table 1) and is 10% above the most recent study. Regionally, our estimates of coastal-erosion-derived carbon flux along the Eurasian coast (12.6 Tg C yr$^{-1}$) agree with earlier studies (4.0–12.8 Tg C yr$^{-1}$)[17], but are 1.8 Tg C yr$^{-1}$ (230%) above the earlier estimate in the Beaufort Sea[36]. The close agreement to the most recent study[16] corroborates our estimate of the nitrogen flux from coastal erosion, which, nevertheless, remains highly uncertain (Fig. 2, methods) due to the large uncertainty in the nitrogen content of coastal soils[28,29], but also because of a potentially large contribution of nitrogen fluxes from subsea coastal erosion[16] that is not accounted for in our Baseline run.

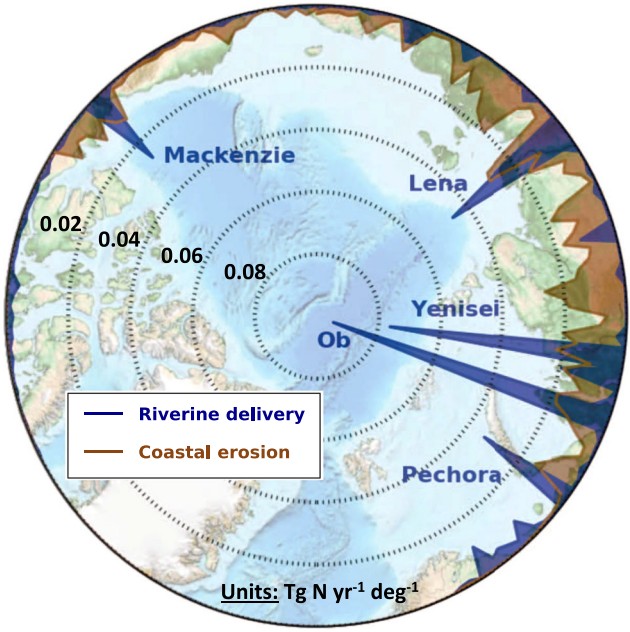

**Fig. 1 Map of annual input of terrigenous nitrogen via rivers and coastal erosion.** Input fluxes are aggregated per degree longitude.

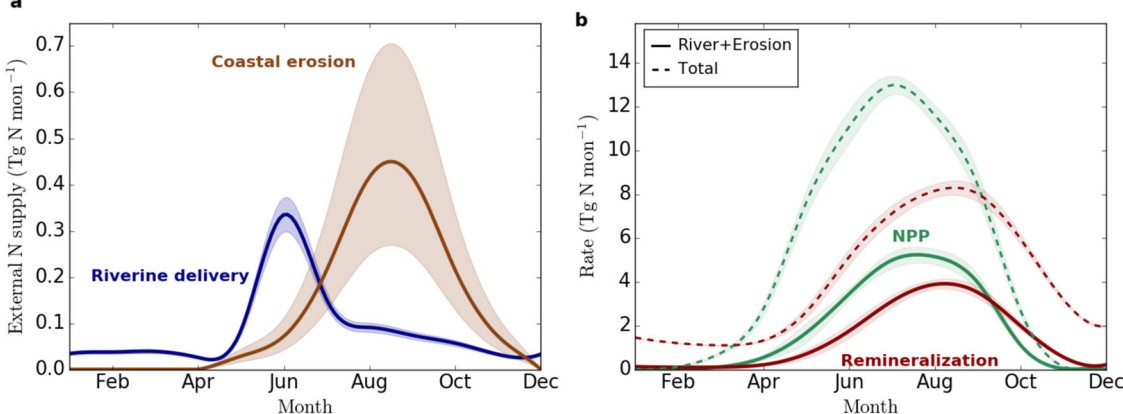

**Fig. 2 Climatology of terrigenous nitrogen input, ocean net primary production and remineralisation in the Arctic Ocean over 2005–2010.** Climatologies of **a** basin-wide terrigenous nitrogen input from rivers (blue) and coastal erosion (brown) and of **b** total basin-wide integrated net primary production (NPP) (green) and pelagic and benthic remineralisation of organic matter (red). Total NPP and remineralisation are indicated as dashed lines (Baseline simulation) while NPP and remineralisation only driven by terrigenous inputs are shown as solid lines (computed as the difference between the Baseline and NoTerr simulations). The envelopes represent **a** uncertainties for the terrigenous nitrogen input (see methods) as well as **b** the simulated interannual standard deviation of NPP and remineralisation over 2005–2010.

**Net primary production driven by terrigenous nitrogen**. The average annual NPP simulated from 2005 to 2010 in the Baseline simulation (including terrigenous nitrogen) is 380 Tg C yr⁻¹ for the Arctic Ocean as defined in this study (Supplementary Fig. 2). When integrated over the larger area north of the Arctic circle, our simulated marine NPP (551 Tg C yr⁻¹) is not significantly different from the remote-sensing-based NPP estimate of 540 ± 25 Tg C yr⁻¹ for the same years[6] (Table 2; Fig. 3a, c). Moreover, simulated NPP agrees within uncertainties with the remote-sensing-based estimate within each regional sea except for the Barents Sea, where the simulated value is 23% lower than the remote-sensing-based estimate, and the Chukchi Sea and the Canadian Arctic Archipelago (CAA), where it is 68% and 59% higher, respectively. These three regions are strongly influenced by inflowing waters from the adjacent oceans. We thus consider this mismatch in NPP to stem from a likely bias in the simulated water inflow, and thus nutrient input, from these oceans[37]. This is further corroborated by these regions not being strongly affected by the terrigenous nutrient input (Table 2; Fig. 3d).

Without terrigenous nitrogen input (NoTerr), the simulated Arctic Ocean NPP is 138 Tg C yr⁻¹ (36%) lower than in the Baseline simulation where this input is considered (Table 2; Fig. 3b). Both coastal erosion and rivers are important nitrogen sources. By comparing the simulation, wherein only the riverine nutrient input is considered (NoCoast), with the baseline and the NoTerr simulations, we find that coastal erosion sustains around 21% (79 Tg C yr⁻¹) of the Arctic Ocean NPP, while rivers sustain around 15% (58 Tg C yr⁻¹).

The relative amount of NPP sustained by terrigenous nutrients is largest on the Siberian shelves: Kara Sea (59%), Laptev Sea (80%) and East-Siberian Sea (57%) (Table 2, Fig. 3b). Leaving out terrigenous input pushes the simulated NPP on the Siberian shelves well below the range of the remote-sensing-based estimates. Furthermore, these shelf seas receive almost no nutrients from adjacent seas, as nitrogen inflow from the Pacific Ocean is diverted to the Central Arctic by the Transpolar Drift before reaching the Laptev Sea[38] and the nitrogen input from the Atlantic Ocean is already consumed in the Barents Sea before reaching the Kara Sea[39]. Conversely, the regional seas that receive considerable lateral nitrogen input from the adjacent oceans (Barents, East-Siberian, Chukchi, and Beaufort Seas) exhibit a

**Table 2 Simulated and remote-sensing-based annual net primary production over 2005–2010[a] north of the Arctic Circle at 66°N [Tg C yr⁻¹] and per region as defined in the remote-sensing-based estimate[6].**

|  | Baseline | NoCoast | NoTerr | Remote-sensing-based |
|---|---|---|---|---|
| Barents Sea | 102 ± 11 | 102 ± 11 | 94 ± 10 | 132 ± 12 |
| Kara Sea | 64 ± 11 | 50 ± 9 | 30 ± 5 | 73 ± 11 |
| Laptev Sea | 59 ± 14 | 21 ± 4 | 11 ± 2 | 56 ± 13 |
| East-Siberian Sea | 48 ± 14 | 27 ± 8 | 19 ± 6 | 44 ± 10 |
| Chukchi Sea | 57 ± 9 | 55 ± 9 | 52 ± 8 | 34 ± 5 |
| Beaufort Sea | 45 ± 10 | 41 ± 9 | 37 ± 8 | 35 ± 4 |
| CAA | 51 ± 6 | 50 ± 6 | 46 ± 5 | 32 ± 3 |
| Greenland Sea | 125 ± 15 | 124 ± 14 | 121 ± 14 | 134 ± 8 |
| Total | 551 ± 89 | 470 ± 69 | 410 ± 59 | 540 ± 25 |
| Arctic Ocean[b] | 380 ± 100 | 301 ± 51 | 243 ± 57 |  |

[a]Uncertainties are consistently calculated as the standard deviation of annual NPP over 2005–2010.
[b]Arctic Ocean is defined in this study (Supplementary Fig. 2) as a subset of "Total" that excludes parts of the regional seas (see Methods).

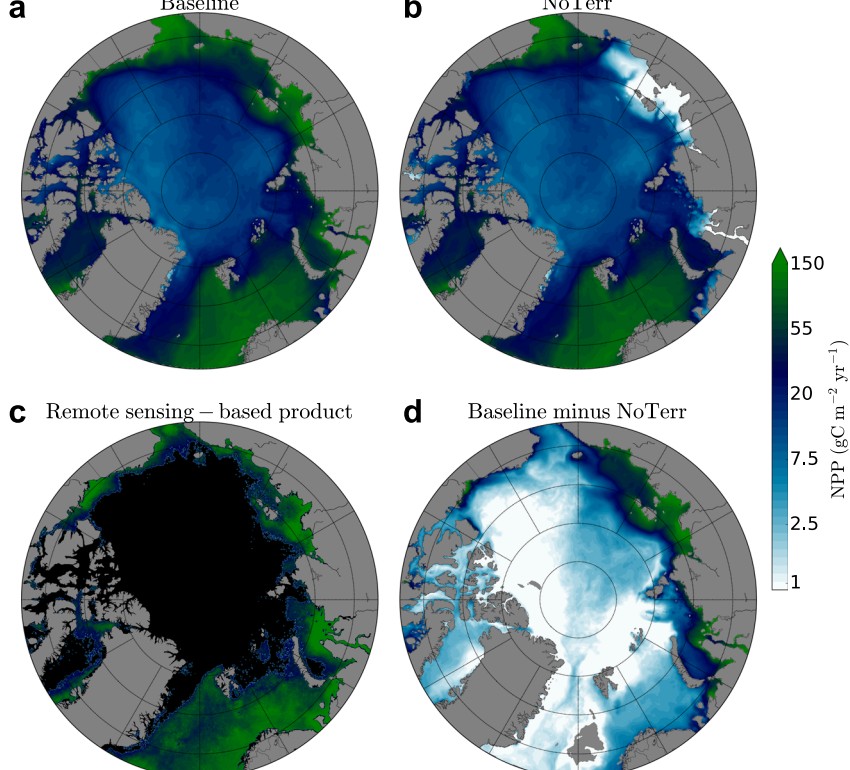

**Fig. 3 Mean annual Arctic Ocean net primary production. a** simulated Arctic Ocean net primary production (NPP) with observation-based nutrient input from rivers and coastal erosion (Baseline), **b** simulated Arctic Ocean NPP without input of terrigenous nitrogen (NoTerr) **c**, remote-sensing-based NPP derived from satellite observations of chlorophyll-a[11] and **d** difference between simulated Arctic Ocean NPP in **a** Baseline and **b** NoTerr simulations.

relatively small, but still visible impact on NPP from terrigenous nitrogen. Temporally, the impact of terrigenous nutrients increases steadily over the summer from 22% in May to 47% in September (Fig. 2).

**Recycling rate of terrigenous nutrients**. The relatively large contribution of terrigenous nitrogen to Arctic Ocean NPP requires a very efficient recycling of the added nitrogen. Assuming a C:N ratio of 122:16, and assuming that every mol of N is used only once by phytoplankton before it is exported, the total annual input of terrigenous nitrogen of 2.6 Tg N yr$^{-1}$ would only support a rise in NPP of 17.0 Tg C yr$^{-1}$, eight times less than the total simulated enhancement in NPP. The much higher stimulation of NPP means that, on average, one mol of terrigenous nitrogen is recycled about seven times before it is exported to the abyss, buried in sediments, or exported laterally to the Pacific or Atlantic Ocean.

This efficient recycling occurs via the remineralisation of marine organic matter in the water column or sediments or via ingestion and excretion by zooplankton. In our simulation, only a small fraction of the marine organic matter produced by the terrigenous nutrients is buried in Arctic sediments (3%) or accumulates as marine dissolved organic matter (3%) and is eventually exported to the adjacent oceans. Instead, most of the organic matter produced by terrigenous nutrients is remineralised, mainly in the sediments (52%) and to a lesser extent within the water column (26%), and recycled via excretion of inorganic nutrients by zooplankton (17%) (Supplementary Fig. 3). Vertically, 88% of the total (water column and sediment) remineralisation occurs at shallow depths (less than 55 m below ocean surface), where around 25% of the Arctic Ocean floor is located (Supplementary Fig. 4).

Benthic processes thus largely contribute to organic matter remineralisation and inorganic nitrogen cycling. Accordingly, the regions with strong NPP and remineralisation are located in very shallow waters where benthic-pelagic coupling is tight (Fig. 3, Supplementary Figs. 2 and 5). Due to the shallow water depth, the cycle of production and remineralisation of organic matter is not dependent on upwelling, as a large fraction of the nutrients is remineralised within the mixed layer. Therefore, we suggest that the large fraction of shallow waters with the associated extensive organic matter remineralisation in sediments of the Arctic shelves support these high pan-Arctic nutrient recycling rates (~7) and thus the relatively large importance of terrigenous nutrients for the Arctic Ocean NPP in this study. Recycling rates from polynyas in the CAA (0.5–3.2)[40–42] are thus not representative for the Arctic Ocean. Hence, their application to estimate pan-Arctic riverine-driven NPP[18] solely from riverine nitrogen input results in an underestimation of pan-Arctic riverine-driven NPP.

**Considering uncertainties**. The results presented here are subject to many uncertainties, in particular associated with missing processes at the land-ocean interface, with the unconstrained reactivity of terrigenous organic matter, with unrepresented terrigenous sources of nitrogen, and with the lack of consideration of the role of terrigenous particulate matter on ocean turbidity. In the following sections, these uncertainties and their potential impact on the amount of nitrogen inputs from land and Arctic NPP are discussed and quantified when possible. This permits us to reassess our model-based estimates of the fraction of the NPP driven by terrigenous sources of nitrogen.

Riverine carbon and nitrogen fluxes are often measured several tens of kilometres upstream from the river mouth[12,18] and thus before the transit through the often complex and biologically active river deltas. In the Arctic, deltas are generally smaller in size than in temperate regions and therefore probably less important for the fate of the riverine matter fluxes, with the exception of the Lena and Mackenzie deltas[43]. In the Mackenzie delta, the concentration of dissolved inorganic nitrogen was found to decrease by 4% across the delta, while that of organic nitrogen increased by 22%, yielding an overall increase in the nitrogen flux[43]. Thus, the alteration of the nutrient fluxes across a river delta is clearly relevant, but likely remains a local effect. Furthermore, we consider our lack of consideration of the impact of the terrigenous particulate matter on turbidity to have only a minimal impact on our results and conclusions. First, turbidity from particles is globally[44] and in the Arctic Ocean[45,46] mostly confined to the very nearshore zone and temporally to the spring breakup, as particulate matter from coastal erosion and rivers settles close to its origin[33,47]. Given the large spatial extent of the Arctic shelves, particles thus only affect a minor part of these shelf seas. Second, while the particles settle quickly out of the euphotic zone, the nutrients remain, unleashing their impact downstream of the river mouth with some delay. Thus, while the consideration of the input of terrigenous particles would result in a spatial and temporal delay of the NPP's response to an input of terrigenous nutrients, we do not expect any fundamental change in the magnitude of the response. Lacking quantitative information on nutrient dynamics in the river delta and on turbidity in the nearshore zone, we do not apply any pan-Arctic adjustment for both effects, following assumptions made in previous studies[12,18–21].

Our assumption of instantaneous remineralisation of organic nutrients into their inorganic form when delivered to the ocean by Arctic rivers and coastal erosion (see Methods) clearly leads to an overestimation of the amount of nutrients available to fuel primary production. Based on observed remineralisation rates of organic matter[48–52] and Arctic Ocean water residence times of freshwater (3.5 ± 2.0 years on Siberian shelves[53], 11 years in the Canada Basin[54]), we estimate that only 60 (20–80)% of riverine dissolved organic nitrogen and 80 (70–90)% of organic nitrogen derived from coastal erosion (see Methods) would be remineralised rapidly enough to fuel Arctic Ocean NPP. Accounting for this "reaction-limited" effect would decrease the input of nitrogen that is available to fuel marine NPP from 1.0 (0.9–1.1) Tg N yr$^{-1}$ to 0.7 (0.5–0.9) Tg N yr$^{-1}$ for riverine fluxes and from 1.6 (1.0–2.5) Tg N yr$^{-1}$ to 1.4 (0.7–2.3) Tg N yr$^{-1}$ for coastal erosion fluxes.

So far, we have not yet accounted for the input of riverine particulate nitrogen nor nitrogen inputs linked to eroding subsea permafrost. Riverine particulate organic nitrogen input to the Arctic Ocean was previously estimated to be around 0.7 Tg N yr$^{-1}$ [22]. However, only around 5–15% of this input leaves the river delta[33] and may contribute to primary production outside the delta. For subsea erosion, the input of organic carbon into the Laptev Sea and the East-Siberian Sea has been estimated at 11 (7–15) Tg C yr$^{-1}$ [16]. Using the same C:N stoichiometric ratio as for Siberian soils and assuming a labile fraction of 70–90% (as for coastal-erosion-derived organic matter) yields a subsea erosion input of organic nitrogen of 1.0 (0.5–1.5) Tg N yr$^{-1}$. Accounting for these two additional terrigenous sources of nitrogen and for the estimated reactivity of organic matter discussed above would yield a nitrogen input that is available to fuel primary production in the Arctic Ocean from rivers of 0.8 (0.6–1.0) Tg N yr$^{-1}$ (−20% compared to prescribed nitrogen input) and from coastal erosion of 2.4 (1.2–3.8) Tg N yr$^{-1}$ (+50%).

**Adjusting the estimate of NPP driven by terrigenous nutrients**. These adjustments in the nutrient delivery ask for a revision of our model-derived estimate of terrestrial-driven Arctic Ocean NPP. In order to determine how much NPP will need to be adjusted, we assume that NPP scales linearly with the amount of

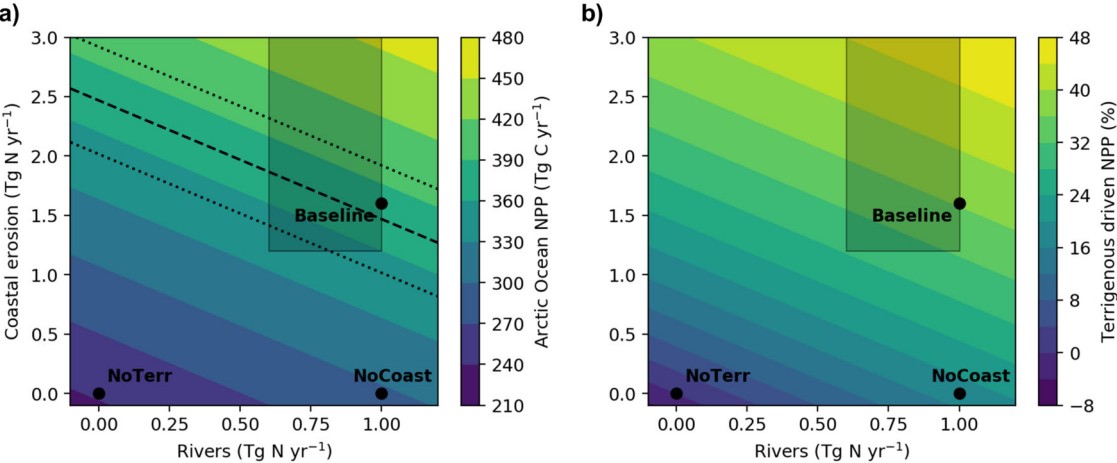

**Fig. 4 Net primary production dependence of terrigenous nitrogen input. a** simulated Arctic Ocean net primary production (NPP) and **b** part of NPP driven by terrigenous nitrogen input from rivers and coastal erosion. Simulated NPP in the Baseline, NoCoast, and NoTerr simulations are shown as black dots. The coloured shading is scaled proportional to the terrigenous nitrogen input. The grey shading indicates the possible range of terrigenous nutrient input available for NPP taking into account the remineralisation rate of terrigenous organic matter and the inclusion of missing sources of terrigenous nitrogen. Remote-sensing-based NPP derived from satellite observations of chlorophyll-a[6] (black dashed line) with uncertainties (dotted black line) are shown in **a** (excluding Nordic Seas and CAA that are mainly out of the area that we defined as the Arctic Ocean).

bioavailable nitrogen supplied by rivers and/or coastal erosion. This linear scaling is corroborated by our three biogeochemical simulations (Baseline, NoCoast, NoTerr), highlighting the N-limited nature of total Arctic Ocean NPP. Using this simple scaling, we adjust the original model-based NPP to 96–255 Tg N $yr^{-1}$, which corresponds to 28–51% of the total Arctic Ocean productivity (Fig. 4). Rivers would thus account for 32–53 Tg N $yr^{-1}$ or 9–11% of total Arctic Ocean NPP, while coastal erosion would account for 64–202 Tg N $yr^{-1}$ or 19–41%. The value at 51% results from a combination of a high nitrogen input and high reactivity and we consider this a likely upper-bound estimate. Such combination would indeed imply low burial efficiency for terrestrial-derived organic matter while several local studies have identified burial as a significant, yet quantitatively uncertain pathway for terrestrial POM in nearshore coastal settings[29,47,55,56]. However, even if we assumed the lowest possible reactivity of organic material and the lowest range of nitrogen input, the terrigenous driven NPP would still contribute at least 28% of Arctic Ocean NPP (Fig. 4). This is indeed a likely lower-bound estimate, since our simulations showed that terrigenous nitrogen is essential to support a NPP that resembles the remote-sensing-based estimates (Figs. 3 and 4).

**Comparison to previous Arctic Ocean NPP studies.** Our assessment suggests a much more prominent imprint of terrestrial inputs compared to previous studies that have estimated the impact of riverine nitrogen on Arctic Ocean NPP. The first study by Tank et al.[18] used fixed recycling rates and estimated that riverine nitrogen sustains 1–4% of Arctic Ocean NPP. Two studies by Le Fouest et al.[19,20] increased this estimate to 9 (5–13)%, in agreement with our river-only estimate (9–11%), by explicitly simulating the nitrogen cycle. As opposed to riverine sustained NPP, no previous estimate exists for Arctic Ocean NPP sustained by coastal erosion. However, our estimates suggest that coastal erosion sustains a significantly larger part of Arctic Ocean NPP than rivers.

**An increasingly complex N-cycle in the Arctic.** Several nonterrigenous sources and sinks of nitrogen in the Arctic Ocean are not considered in this study, e.g. nitrogen input from melting

Greenland glaciers, atmospheric deposition, di-nitrogen fixation, and nitrogen losses through benthic denitrification. The inorganic nitrogen flux from Greenland glaciers is estimated to be of the same order of magnitude as riverine inorganic nitrogen, e.g. 0.3 Tg N $yr^{-1}$ [57]. However, this flux is mainly directed into the CAA, the Nordic Seas, and the Labrador Seas from where they flow southward and not northward into the Arctic Ocean. Atmospheric deposition of nitrogen is estimated at approximately 0.1 Tg N $yr^{-1}$ over the Arctic Ocean area without perennial sea-ice cover[58] and is thus small compared to terrigenous nitrogen inputs. Similarly, nitrogen fixation has long been believed to be negligible in the Arctic Ocean due to low temperatures. However, recent studies in the Western Arctic challenge this view[59,60] and estimate a maximum potential for Arctic Ocean nitrogen fixation of 3.5–9.2 Tg N $yr^{-1}$ if this process was similar in all ice-free Arctic seas[60]. However, this potential pan-Arctic nitrogen fixation flux is of the same order as denitrification in Arctic sediments (2.8–29.0 Tg N $yr^{-1}$)[61,62], which is not represented in the model either. Thus, not simulating nitrogen fixation might offset the effect of not simulating denitrification in sediments, although we recognise that these two processes might be partly decoupled in space and time. However, the remarkably good spatial agreement between remote-sensing-based and simulated NPP (Fig. 3) suggests that, despite not taking into account all nonterrigenous sources and sinks of nitrogen, our model set-up represents the Arctic Ocean nitrogen cycle and NPP rather well.

In conclusion, this study provides a combined estimate of terrigenous nutrient fluxes into the Arctic from rivers and coastal erosion resolved in space and time and an estimate of the impact of these fluxes on Arctic Ocean NPP. Our finding that nutrient fluxes from coastal erosion are likely larger than riverine fluxes supports the previously hypothesised importance of coastal erosion for Arctic Ocean NPP and the dependent ecosystem[13]. However, the still large uncertainties with respect to the terrigenous fluxes and their respective lability highlight that more research is needed to better quantify the individual components of the Arctic Ocean nitrogen budget, in particular at the land-sea interface.

Despite all uncertainties, our results indicate that terrigenous nitrogen fluxes sustain 28–51% of Arctic Ocean NPP and suggest that coastal erosion is one of the main drivers of the

Arctic Ocean NPP. Therefore, increases in Arctic Ocean NPP over the last decades[6,7,63] that were exclusively attributed to decreasing sea-ice extent, a longer growing season, and ocean circulation changes may as well be partly caused by increases in riverine discharge[64] and coastal erosion[13,30]. Moreover, terrigenous nitrogen input will likely increase over the 21st century[13,16,65] and thus further increase Arctic Ocean NPP[14]. It is therefore of great importance that the terrigenous nutrient fluxes are better constrained and that they are consistently implemented in Earth System Models to improve the highly uncertain projections of Arctic Ocean NPP[8] and the associated fishery catch potential over the 21st century[9].

## Methods

**Arctic Ocean.** In this study, we define the Arctic Ocean with the Fram Strait, the Barents Sea Opening, the Bering Strait, and the Baffin Bay as boundaries (Supplementary Fig. 2)[66].

**Pan-Arctic extrapolation of river flux measurements.** We derived a new forcing file representing a climatology of average monthly river fluxes of alkalinity (here assumed to represent entirely carbonate alkalinity, $A_C$), dissolved inorganic carbon ($C_T$), dissolved organic carbon (DOC), dissolved inorganic nitrogen (DIN), dissolved organic nitrogen (DON), total dissolved phosphorus (TDP), and silicic acid ($Si_T$). This dataset is consistent with monthly data of coastal river discharge[67]. To do so, the observation-based flux estimates determined over the first decade of the 21st century from the six largest Arctic rivers (Mackenzie, Yukon, Kolyma, Lena, Ob, Yenisei) from the Arctic Great River Observatory (ArcticGRO) dataset[12,21] were extrapolated to all river watersheds draining to coastal stretches north of 60°N latitude, using the STN30p stream network[68] for delineation. Although the Yukon river does not drain directly into the Arctic Ocean, its large drainage basin lies predominantly north of 60°N and is representative of other ungauged Arctic rivers that drain directly into the Arctic Ocean.

First, the annual fluxes were spatially extrapolated to obtain an optimised estimate with regard to the total annual fluxes. Then, the annual fluxes were redistributed over the seasonal cycle using an empirical approach. By predicting the average annual fluxes (prediction of spatial variability only) first, these fluxes have lower uncertainty in the flux predictions than if spatial and seasonal variability had been predicted at the same time. Note that for the N-budget and NPP calculations for the Arctic Ocean, the mean average annual riverine input fluxes are of highest importance, and it is thus our priority to make the predicted mean annual fluxes as reliable as possible. The seasonal distribution of these fluxes over the year are of subordinate importance.

To identify potential predictors of the average annual fluxes of carbon and nutrients to the coast, a variety of catchment properties from available geodata were calculated (Supplementary Table 1 and Supplementary Fig. 6). Multiple linear regression was applied to extrapolate the annual river fluxes of the different carbon and nutrient species. For each of the carbon and nutrient species, the four predictors which best explained the differences between the six largest Arctic rivers, i.e. which gave the lowest root mean squared error (RMSE) between observed and predicted fluxes, were identified (Supplementary Table 2).

For $C_T$ and $A_C$ fluxes statistical models based on the same dataset of observed fluxes were previously developed by Tank et al.[21] and use as predictors the carbonate index, the areal extent of permafrost, the areal proportion of glaciers, and runoff. We decided to use the same statistical models and geodata sets of predictors as Tank et al., but refitted the equations to account for slightly different river basin averages of the predictors that we calculated. While observed annual runoff from the six rivers was used to fit the regression, a global runoff dataset[69] was used to extrapolate $C_T$ and $A_C$ fluxes to all boreal and Arctic rivers on the Northern Hemisphere following Tank et al. In order to obtain fluvial $C_T$ and $A_C$ fluxes that are consistent with the river discharge used as forcing data in the ocean-biogeochemical model NEMO-PISCES[67], $C_T$ and $A_C$ concentrations were calculated by dividing the extrapolated annual fluxes by the annual runoff[69]. These concentrations were then multiplied with the here used river discharge[67].

For DOC and dissolved nutrients, for which no statistical models existed beforehand, multiple linear regression was used to predict concentrations directly, testing all possible combinations of four predictors (Supplementary Table 2). For DOC, DON, DIN and TDP, the retained predictors were the areal extent of permafrost, the areal proportion of lakes, as well as the average clay and organic carbon contents of the topsoil. To avoid unreasonable numbers for extrapolated carbon and nutrient concentrations, the maximum and minimum flux weighted annual concentrations from the ArcticGRO data were imposed as upper and lower bounds for our extrapolation (Supplementary Table 3). The so-obtained concentrations were then multiplied by the river discharge used as forcing data in the ocean-biogeochemical model PISCES[67] to obtain DOC and dissolved riverine nutrients fluxes.

In a next step, the average monthly fluxes were estimated based on the empirical relationship between the relative seasonal variations in matter fluxes and in river discharge:

$$F_{\text{monthly}} = F_{\text{annual}} a_1 \left( \frac{Q_{\text{monthly}}}{Q_{\text{annual}}} \right)^{a_2} \tag{1}$$

with $F_{\text{monthly}}$ and $F_{\text{annual}}$ being the monthly and annual flux, $Q_{\text{monthly}}$ and $Q_{\text{annual}}$ being the monthly and annual discharge, and $a_1$ and $a_2$ being fitted parameters (Supplementary Table 4). Only for DIN, the position of each month in the seasonal cycle was used as an additional predictor:

$$F_{\text{monthly}} = F_{\text{annual}} \left[ a_1 \left( \frac{Q_{\text{monthly}}}{Q_{\text{annual}}} \right)^{a_2} + b_1 \sin \left( \frac{b_2 + \text{month}}{6} \pi \right) \right] \tag{2}$$

with $b_1$ and $b_2$ being additional fitting parameters (Supplementary Table 4). For species other than DIN, no statistically significant ($p < 0.05$) fit for this part of the equation could be obtained.

Note that the exponent $a_2$ describes how concentrations in carbon and nutrients react to changes in discharge, with $a_2 < 1$ indicating a decrease in concentrations with increasing discharge, i.e. a dilution effect, and $a_2 > 1$ indicating an increase in concentrations with discharge, i.e. the so-called "flushing effect". For $C_T$, $A_C$, silica, and DIN the fitted $a_2$ values indicate a dilution effect, consistent with observations[12,21]. Accordingly, we predict highest DIN concentrations during winter, when discharge is lowest (winter baseflow). The additional seasonal component which we fitted for monthly DIN fluxes represents an additional decrease in DIN fluxes over summer, which is as well consistent with the finding that during summertime, primary production additionally decreases riverine DIN fluxes[12]. For DOC and DON, our fitted $a_2$ values indicate a flushing effect, which means that the increase in river flow during spring/summer contributes overproportionally to the annual riverine DOC and DON exports, again consistent with observations[12].

To avoid unrealistic $A_C$ to $C_T$ ratios, we derived minimum and maximum observed ratios from the ArcticGRO data and imposed them on our monthly estimates of $A_C$. The minimum and maximum possible $A_C$ (µeq $L^{-1}$) to $C_T$ (µmol $L^{-1}$) ratios were thus defined as 0.24 and 1.07, respectively. Finally, to obtain the best possible forcing dataset of fluvial matter fluxes to coastal waters, the observation-based monthly average concentrations from the ArcticGRO data were used to calculate monthly fluxes for the six largest Arctic rivers. Thus, the extrapolated concentrations were only used for the ungauged part of the Arctic watershed. The final forcing file is provided on a regular 1° grid.

**Calculation of pan-Arctic carbon and nutrient fluxes from coastal erosion.** We derived a new forcing file representing a climatology of average monthly total organic carbon (TOC), total nitrogen (TN), and total phosphorus (TP) fluxes from coastal erosion to the Arctic Ocean. First, TOC fluxes from coastal erosion were calculated by multiplying spatially resolved estimates of coastal erosion rates (based on observations from 1950 to 2010) by estimates of carbon content in coastal soils[30]. Then nitrogen fluxes were estimated by assuming a stoichiometric C:N ratio of 15.1:1.0 for the North American coast[29] and 10.5:1.0 for the Eurasian coast[28]. Eventually, the phosphorus fluxes were calculated based on a global estimate of the soil N:P ratio of 13.1:1[31].

The obtained annual fluxes were then seasonally divided (2% in May, 5% in June, 15% in July, 29% in August and September, 15% in October, 5% in November) based on observations of coastal erosion rates[70–73]. The final forcing file is provided on a regular 1° grid, consistent with the forcing of riverine carbon and nutrient inputs.

**Ocean-biogeochemical model NEMO-PISCES.** To analyse the effect of the riverine delivery of carbon and nutrients on the Arctic Ocean biogeochemistry, we used the version 3.2 of the ocean modelling framework 'Nucleus for European Modelling of the Ocean' (NEMO). This framework includes the ocean dynamics part OPA[74], the version 2 of the Louvain-La-Neuve sea Ice Model LIM[75] and the 'Pelagic Interaction Scheme for Carbon and Ecosystem Studies' (PISCES) biogeochemical model[32]. The model was used with the eddy admitting DRAKKAR configuration ORCA025[76].

The biogeochemical model PISCES[32] simulates the cycles of dissolved inorganic and organic carbon, total alkalinity, oxygen, nitrate ($NO_3^-$), ammonium ($NH_4^+$), dissolved inorganic phosphate, silicic acid and iron. In addition, four living pools are simulated: nanophytoplankton and diatoms, and microzooplankton and mesozooplankton. The phytoplankton growth depends on temperature and is limited by light and nutrient availability. The photic depth reduces with increasing phytoplankton due to shading, whereas a possible shading from particles is not accounted for in the model. The C:N:P ratio of 122:16:1[77] is prescribed in all living and nonliving organic compartments of PISCES. In addition to the living compartments, PISCES simulates semi-labile dissolved organic matter (DOM), and small and big sinking organic particles.

External nutrient sources to the Arctic Ocean water column comprise lateral inflow from adjacent oceans, sediment remineralisation, river fluxes, and coastal erosion. The nutrient fluxes exchanged between the Arctic and the neighbouring oceans were calculated dynamically as the global model configuration was used. Sediment remineralisation is also calculated dynamically within the model. Nutrient fluxes from rivers and coastal erosion were calculated from the newly

generated monthly 2-D forcing files. The fluxes from the regular 1° grid were rescaled to the curvilinear model grid. To do so, each model grid cell was assigned to its nearest grid cell on the 1° grid. The flux from the 1° grid was then divided between the assigned model grid cells, for coastal erosion in proportion to the surface area of each cell and for rivers in proportion to the runoff in each cell. This approach ensures the conservation of mass of the calculated fluxes and correlates runoff with carbon and nutrient fluxes. Vertically, the river input was divided on the first two vertical model grid levels (0–13 m) to account for a finite river depth.

**Land-ocean interface**. All terrigenous fluxes were added to the respective inorganic ocean model variable (nitrogen was added in the form of nitrate), as the standard version of PISCES does not explicitly simulate DOC, DON and DOP separately. Instead, DON, DOP and DOC are simulated by a single tracer for semi-labile organic matter assuming a marine stoichiometric C:N:P ratio of 122:16:1. As opposed to this marine ratio, the stoichiometric C:N:P ratio in our estimated Arctic river fluxes is ~1188:25:1[12,21] and the C:N:P ratio in fluxes from coastal erosion is ~147:13:1[28,29,31]. The configuration of PISCES does not allow to mix organic matter with different Redfield ratios. In order to conserve the total fluxes of carbon and nutrients, we chose to add all terrigenous organic matter fluxes to the ocean in their inorganic form, which is an overestimation that we quantify in the "Uncertainty estimation" section below.

**Simulation strategy**. In this study, we ran three global simulations with the NEMO-PISCES model from 1990 to 2010: (1) one with the newly derived forcing dataset for terrigenous carbon and nutrient fluxes from rivers and coastal erosion (Referred to as Baseline), (2) one with coastal erosion fluxes set to zero (NoCoast), (3) one with coastal erosion fluxes and riverine carbon and nutrient fluxes set to zero (NoTerr). All three simulations were initialised in 1990 with the output from previously published simulations[37] and ran with the same model parameters and external forcing, apart from the different input of terrigenous carbon and nutrients in the Arctic Ocean. The initialisation in 1990 from a simulation with different terrigenous carbon and nutrient input leads to a transient period that is visible in the annual Arctic Ocean NPP from 1990 to 2004 (Supplementary Fig. 7). Only model output after the transient period from 2005 to 2010 was analysed to ensure that the results are solely driven by the new terrigenous carbon and nutrient inputs.

**Uncertainty estimation**. In this study, the uncertainties related to the quality of the forcing files and the uncertainties related to the lability of the terrigenous organic matter are separately quantified.

The estimated terrigenous carbon and nutrient fluxes have uncertainties due to the scarcity of the data in the difficult to sample Arctic environment and due to the extrapolation on the pan-Arctic scale. These uncertainties had to be estimated here because previously published observation-based river fluxes[12,21] and erosion rates[30] were given without uncertainties. The uncertainties for riverine carbon and nutrient fluxes arise mainly from the temporal and spatial extrapolation[12]. The uncertainty from temporal extrapolation was previously only estimated for $C_T$ fluxes (4%)[21] and we assumed that this uncertainty is representative for DOC and nutrient fluxes[12], too. To quantify the uncertainty due to the spatial extrapolation of the riverine fluxes, we relied on the range of previously reported extrapolations of riverine DOC exports to the Arctic watershed (25–30 Tg C yr$^{-1}$)[12,27,78], which also includes our own basin-wide annual riverine DOC flux of 27.6 Tg C yr$^{-1}$. The uncertainty in both directions is thus estimated at about 10%, which combined with a temporal uncertainty of 4%, leads to a total, symmetric uncertainty for the riverine fluxes of carbon and nutrients of ±11%.

In comparison to river fluxes, the fluxes from coastal erosion have much higher uncertainties, mainly caused by uncertainties in the erosion rate in space and time, the soil organic carbon content, the C:N:P ratio in the soil, and the extrapolation of these measured quantities. Both, the previously published spatially extrapolated erosion rates and soil organic carbon content of the eroded material are reported without uncertainties[30]. In particular the soil carbon content varies strongly and has local maxima, which are often missed by low-resolution sampling[79]. Based on these spatial variations with localised maxima, we assume an asymmetric uncertainty for carbon coastal erosion fluxes of 30% towards lower values and 50% towards higher values. For nutrient fluxes, the C:N:P ratio adds another source of uncertainty. To constrain this uncertainty, we use the standard deviation of the C:N:P ratio among existing measurements, which is 27% for the C:N ratio[28,29] and 6% for the P:N ratio, which is derived from global estimates[31]. We thus obtain a relative uncertainty range from −40 to +57% for organic N fluxes and −41 to +57% for organic P fluxes from coastal erosion, which are larger than for the riverine fluxes.

The lability of the organic matter flux adds another source of uncertainty, which is assessed in the following way. For rivers, estimates of lability of riverine organic matter in the Arctic Ocean vary strongly from 20–40% of riverine DOC from Alaskan rivers being remineralised within three months[24], over 50% of terrigenous DOC on the Siberian shelf being remineralised within a year[49], to 62–76% of riverine DON on the Siberian shelf being remineralised within in couple of months[50]. Taking into account that residence times on the Siberian shelves reach 3.5 ± 2 years[53] and even 10 years in the Canada basin[54], it was estimated that a large fraction of the terrigenous organic carbon will be remineralised within the

Arctic Ocean, in agreement with the finding that only 21–34% of the riverine organic carbon is leaving the Arctic Ocean[48]. Our study thus likely overestimates the amount of the NPP fuelling, labile DON pool. Based on the available studies, we thus estimate 20–80% of the DON flux to be labile.

The organic matter from coastal erosion is expected to be even more reactive than riverine supplies, in agreement with the findings that thawing permafrost soils release one of the most labile forms of organic matter in nature[80–82]. Observation-based estimates of remineralisation rates of terrigenous organic matter in the Laptev Sea and East-Siberian Sea are 0.87 Tg N yr$^{-1}$ in sediments[52] and 0.38 Tg N yr$^{-1}$ in the water column[51]. Scaled up to the entire Arctic Ocean, using the primary production sustained by terrigenous nutrients as scaling factor, remineralisation rates of 1.56 Tg N yr$^{-1}$ in sediments and 0.68 Tg N yr$^{-1}$ in the water column are obtained. Compared to our estimated input of organic nitrogen from rivers and coastal erosion (2.37 Tg N yr$^{-1}$), this yields a remineralisation rate of 95%. The almost complete remineralisation of terrigenous organic matter contradicts the observations of strong, but quantitatively uncertain sedimentation in the coastal Arctic[29,47,55,56]. The apparent mismatch between organic matter input and remineralisation might be due to missing sources of terrigenous matter in our study, e.g. subsea coastal erosion[16] or the riverine particulate matter[22] that does not precipitate in the river delta[33] or due to resuspension of terrestrial organic matter in the nearshore zone[34,56]. Following the observed remineralisation rates for terrigenous organic carbon[51,52] in sediments and the water column, we estimate that 80 (70–90)% of our estimated input of organic nutrients released by coastal erosion will contribute to NPP before leaving the shelf seas.

**Remote-sensing-based NPP estimates**. NPP was compared to estimates derived from remotely sensed ocean color from 2005 to 2010[6,11]. Remote-sensing-based NPP is estimated from satellite observations of chlorophyll using an algorithm that was calibrated with in-situ observations of chlorophyll-a. The reported uncertainties (Table 2) are calculated as the standard deviation of annual NPP from 2005 to 2010.

## Data availability
The spatially resolved input of nitrogen from coastal erosions and rivers and the model output that supports the findings of this study are available under https://doi.org/10.17882/76983.

## Code availability
The model code is publicly available at https://www.nemo-ocean.eu/.

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

## Acknowledgements

This study has received funding from the European Union's Horizon 2020 research and innovation programme under grant agreement No 643052 (C-CASCADES), No 641816 (CRESCENDO), No 776810 (VERIFY) and No 821003 (4 C). The work reflects only the authors' view; the European Commission and their executive agency are not responsible for any use that may be made of the information the work contains. This study has also received funding from the Agence Nationale de la Recherche grant ANR-18-ERC2-0001-01 (CONVINCE), the MTES/FRB Acidoscope project and the ENS-Chanel research chair. We also thank C. Ethé for help with the NEMO-PISCES simulations.

## Author contributions

J.T. and R.L. led the study. R.L. created the forcing files for terrigenous carbon and nutrients and J.T. the ocean-biogeochemical simulations. Model analysis and production of figures was done by J.T. with help from R.L., P.R., N.G. and L.B. The manuscript was drafted by J.T. and modified and improved by R.L., P.R., N.G. and L.B.

## Competing interests

The authors declare no competing interests.
