## [Peer Review File · Nature Communications]

REVIEWERS' COMMENTS<

Reviewer #1 (Remarks to the Author):

The manuscript provided by Terhaar et al. represents a revised version of an earlier manuscript submitted to another journal in the Nature Publishing group, which was then transferred along with the reviews to NatComms.

I had reviewed this earlier version so that I will not add a new general paragraph on the overarching importance and potential impact of the manuscript. Instead, I have copied and slightly revised these general remarks below since they are still (and even more now) valid from my point of view.

The manuscript by Terhaar et al. entitled "Around one third of current Arctic Ocean primary production sustained by rivers and coastal erosion" is of great interest for the geoscientific and climate change community in the Arctic and beyond. The manuscript seeks to close a huge knowledge gap which is the contribution of Arctic rivers and coastal erosion to the Arctic nutrient budget and ultimately to the Arctic food web and carbon/nutrient cycle. Therefore, I find this manuscript suitable for the chosen journal. The authors provide clear and strong evidence for their conclusions. The topic is of utmost importance to any scientist working on modern and future-directed climate research in the Arctic. Since the Arctic is in the epicenter of (current and future) climate warming and plays an important role in the global climate and ecological system, this study will find a large readership beyond Arctic research and will be well-received also beyond the scientific community.

The manuscript makes the very interesting point that net primary production (NPP) in the Arctic Ocean and especially along the broad Arctic shelves is mainly driven by nutrient input by rivers and coastal erosion. The authors use observations from the six largest rivers and measured coastal erosion rates, to upscale and to construct a pan-Arctic, spatio-temporally resolved estimate of the terrigenous input of carbon and nutrients to the Arctic Ocean. These estimates then feed into an ocean-biogeochemical model that verifies the hypothesis that current annual Arctic Ocean NPP is driven by terrigenous nutrient inputs, with the largest contribution stemming from coastal erosion. Especially the latter part has made its way into the literature as a hypothesis over the last years but could not be treated adequately without a large-scale ocean-biogeochemical model. With this study we get the first spatially- and seasonally-resolved estimate of terrigenous nutrient fluxes from rivers and coastal erosion which is based on actual data. Moreover, it is the first study that estimates the impact of nutrients from coastal erosion on the Arctic Ocean NPP.

I am confident that this study will foster more research and model development of the Arctic carbon and nutrient budgets under climate change scenarios (decreasing sea ice extent, longer thaw periods, changes in river discharge and rates of coastal erosion), especially with all the uncertainties that remain. It is of great importance that the terrigenous carbon and nutrient fluxes in the Arctic are implemented in Earth System Models to improve the highly uncertain projections.

The results are convincing and the methods are sound. Statistical methods and uncertainties are made available. The authors treat caveats and uncertainties of data and statistics in a very transparent way, which is highly acknowledged.

In the current version of the manuscript, the author team has greatly improved the clarity and potential impact of the manuscript. The authors have done a very good job at all parts of the manuscript and supplementary files as far as I can judge. They have addressed all of my previous concerns to my satisfaction. As far as I can judge the other reviewer's comments and suggestions, the authors have tried their best to incorporate all three views. However, I am not a modelling expert, so that other experts need to follow up on that.

I have left some minor comments and questions directly at the annotated manuscript and supplement. They should be easy to address and are mostly of technical nature. I suggest the manuscript to be

considered for acceptance after minor revisions.

Michael Fritz (Alfred Wegener Institute, Helmholtz Centre for Polar and Marine Research)

Reviewer #2 (Remarks to the Author):

The authors have invested a great deal of time and effort to address the questions raised by all three reviewers and for that I think the manuscript is much improved. In particular, I appreciated the clarification around the inputs of DIN and DON and the added text surrounding the land-ocean interface. As well, the now included text about how the model approaches the fate of nutrients from rivers and coastal erosion and the impact of the uncertainty of the organic matter flux gives the reader a more comprehensive view of the model output and its limitations. This is nicely done. I'd like to add a few comments for the authors to consider to follow-up on their responses to my initial questions (below, see revision reply) and make some further suggestions for minor revisions (below, see minor suggestions). After these are considered, I support the acceptance of this article for publication in Nature Communications.

Revision reply:

Comment 3.3: I still find the use of the term "data-based" quite confusing in the text, as it appears to be used interchangeably for in situ measurements and remotely sensed data, two types of data which I think should be distinguished when considering their use within (or in validation of) the model. I'd like to suggest that "data-based" be replaced with "remote sensing-based" where it is referring to satellite observations throughout the text; and replaced with "observation-based" (or another distinguishing term) where it is referring to extrapolations based on measured values. While I agree that these inputs are all observations and are all "data", I think it is important that the reader be able to distinguish extrapolations based on in situ measured values (as your climatologies are) and those determined by proxy (satellite remote-sensing products). Please see my suggestions below (see minor suggestions) on specific instances where confusion arises.

Comment 3.13,3.17: I'm still concerned that the importance of coastal erosion is determined from a system where turbidity is not accounted for. To me this means that while you may have stimulated primary productivity by the nutrients released from coastal erosion in the model, in the real ocean the turbidity associated with coastal erosion may be too high to permit enough solar radiation for primary production to occur. This moves an erosion dominated coastline from N-limitation to light limitation; which would not be accounted for in your model, producing an over-estimate of primary productivity. I think some more discussion on the implications of not accounting for turbidity is warranted.

Comment 3.18: The authors address my original question about the extrapolation of total annual fluxes to determine seasonal fluxes quite thoughtfully, I think it would benefit the paper to include some details of this explanation in the methods.

Minor suggestions for the main text and supplement include:

line 49, suggest starting the sentence with "It has been estimated that..." as the studies reported are remote sensing-based observations and are therefore not accounting for changes throughout the water column

line 71, suggest replacing "exports" with "composition" as the exports are calculated whereas it is concentrations that are measured

line 72, there have been time series observations carried out in several medium and small Arctic

rivers, I think they should be mentioned here, including:

Li Yung Lung, J. Y. S., Tank, S. E., Spence, C., Yang, D., Bonsal, B., McClelland, J. W., & Holmes, R. M. (2018). Seasonal and geographic variation in dissolved carbon biogeochemistry of rivers draining to the Canadian Arctic Ocean and Hudson Bay. *Journal of Geophysical Research: Biogeosciences*, 123, 3371–3386. <https://doi.org/10.1029/2018JG004659>

McClelland, J. W., Townsend-Small, A., Holmes, R. M., Pan, F., Stieglitz, M., Khosh, M., & Peterson, B. J. (2014). River export of nutrients and organic matter from the North Slope of Alaska to the Beaufort Sea. *Water Resources Research*, 50, 1823–1839. <https://doi.org/10.1002/2013WR014722>

Holmes, R. M., Peterson, B. J., Gordeev, V. V., Zhulidov, A. V., Meybeck, M., Lammers, R. B., & Vörösmarty, C. J. (2000). Flux of nutrients from Russian rivers to the Arctic Ocean: Can we establish a baseline against which to judge future changes? *Water Resources Research*, 36(8), 2309. <http://doi.org/10.1029/2000WR900099>

Holmes, R. M., McClelland, J. W., Raymond, P. A., Frazer, B. B., Peterson, B. J., & Stieglitz, M. (2008). Lability of DOC transported by Alaskan rivers to the Arctic Ocean. *Geophysical Research Letters*, 35, L03402. <https://doi.org/10.1029/2007GL032837>

I suggest modifying the sentence beginning at line 71 to something along the lines of: "Measurements of riverine nutrient composition are taken frequently at the six largest Arctic rivers (Mackenzie, Yukon, Kolyma, Lena, Ob, Yenisei), and have been periodically recorded in several smaller river systems (e.g., Holmes et al., 2000, 2008; McClelland et al., 2014; Li Yung Lung et al., 2018), but few time series observations have been collected elsewhere."

line 74, suggest modifying the sentence to read, "...fluxes from the six largest rivers were extrapolated..."

line 75, suggest modifying the sentence to read, "Riverine nutrient inputs determined from previously published pan-Arctic organic carbon fluxes have been used to estimate riverine-driven NPP to vary between 4% and 10%..."

line 84, as indicated in my previous review, the term "data-based" as used in the text is still very confusing. I suggest "data-based" be replaced with "remote sensing-based" where it is referring to satellite observations throughout the text; and replaced with "observation-based" (or another distinguishing term) where it is referring to extrapolations based on measured values. Specifically I would suggest:

line 84, replace "spatially resolved, data-based forcing set" with "spatially resolved forcing derived from in situ data, set"

in Table 2 title, at line 190, it is not clear here what "data-based" refers to, are the simulated values not determined based on the Arctic-GRO measured data? Perhaps just remove "and data-based" in Table 2, final column of Table (right-hand column), this instance is referring to "remote sensing-based" correct?

in Figure 3, title to figure 3c, and in caption (line 204), replace with "remote sensing-based";

in Figure 3, caption (line 202) I believe refers to the Arctic-GRO data, here I would suggest simplifying to "simulated Arctic Ocean NPP with nutrient input..." as it is clear your simulation is using the Arctic-GRO data

line 221, here I believe "observation based" is also referring to "remote sensing-based" estimates? (see why it's very confusing?)

line 320, replace "data-based" with "remote sensing-based"

Figure 4, caption (line 329/330) replace "data-based" with "remote sensing-based"

line 366, replace "data-based" with "remote sensing-based"

line 403, here, again, "data-based" refers to measured values, here I would just remove "data-based" completely

line 608, here "Data-based" can be left as is since it's referring to ALL the estimates based on data
line 610, here "observation based" is referring to "remote sensing-based" and should be changed
Supplement. Figure S5, line 43, replace "data-based" with "remote sensing-based"

line 86, "(in terms of basin size)" this is confusing, in what ways are they not the six largest rivers?

line 86, suggest including the caveat, "...Arctic rivers^{12,21}, five of which drain directly into the Arctic Ocean, which..."

line 87, suggest "using a spatially"

line 95, here it suggests you do use canonical Redfield ratios (106:16:1)? if not, you should state the ratios used here. I agree with the other reviewer that if you are not using Redfield ratios explicitly (C:N:P = 106:16:1) then you should not refer to them as such. You can simply use the term "stoichiometric ratios" or "C:N:P stoichiometry" in place of Redfield references.

line 112, why is the Pechora River now part of the results? From the methods I understood that only data from the six largest Arctic rivers were included?

Figure 2, please note spelling in Fig 2b "remineralization" versus the caption "remineralisation"

line 143, suggest "six largest Arctic rivers"

line 163, suggest "The estimate presented here of"

line 168, suggest removing the word "earlier"

line 196, why do the (No Coast) data not appear in Table 2? I think these should be included (even if they are just determined by a subtraction)

line 224, suggest removing "Redfield" as "Assuming a C:N ratio of 122:16" is sufficient

line 250, what are the pan-Arctic nutrient recycling rates you determine? it would be nice to have some ranges to compare to the literature values in the next sentence

line 282, is 60% intended to be the average of 20-80%? (if so, it is incorrect)

line 283, does "organic matter" refer to C or N or both?

Figure 4, caption line 324, suggest remove "in dependence nitrogen" this looks like a typo

line 340, this is not actually changed as indicated in the response to reviewer file (see response 3.10), this sentence should begin, "The first study by Tank et al.,¹⁸..."

line 344/345, again, I'm concerned that these coastal erosion numbers derive from a system where turbidity is not accounted for (see notes above). I think this should be included as a possible area of uncertainty.

line 374, suggest "coastal erosion on Arctic Ocean"

line 377, it might be worth including a note that coastal erosion is limited in extent and has its main impact immediately adjacent to the coast

line 404, suggest "...estimates determined over the first decade of the 21st century from the six

largest Arctic rivers..."

line 407, I think a caveat about the Yukon River is warranted here, as I do think it should be included in your data set (it's important for scaling), but as the other reviewer noted it does not drain into the AO directly. I suggest adding something like the following at line 407, "We note that although the Yukon River does not drain directly into the Arctic Ocean, its large drainage basin lies predominantly north of 60N and is representative of other un-gauged Arctic regions that drain directly into the Arctic Ocean..."

line 416, suggest "six largest Arctic rivers"

line 422, I think more explanation is needed here as to why a global runoff data set was used to extrapolate Ct and Ac fluxes, especially as this was not done for other nutrients

line 451, suggest replace "in consistency" with "consistent"

line 458, suggest replace "in consistency" with "consistent"

line 465, suggest "six largest Arctic rivers", I've made this suggestion a few places as reading "six big rivers" does not indicate that these rivers are the largest, you could use "big-6" as is done in Holmes et al.,'s work, but I think saying "six largest Arctic rivers" is more appropriate

line 467, I hadn't thought of this until noticing the inclusion of the Pechora River earlier, but how do you determine WHERE along the Arctic coast these inputs occur? (and does it matter?) For example, are the inputs from all ungauged areas in NA distributed evenly across the coast or is there one point source (like the Mackenzie) where these materials are brought into the model ocean? Perhaps just adding a line or two about how NEMO-PISCES treats these additions.

line 498, here you include a note about possible shading effects of particles, as above, I think some more thought to the implications is warranted in the discussion

line 498, suggest remove "Redfield"

line 522, is this the marine stoichiometric ratio you refer to below? suggest "a marine stoichiometric C:N:P ratio"

line 523, suggest remove "Redfield"

line 526, does PISCES use the canonical Redfield ratio (106:16:1) or the marine ratio listed above (122:16:1) from Takahashi? If it uses Redfield then you can leave this in, however if not you could replace "Redfield" with "stoichiometric"

line 550, suggest replace "hostile" with "difficult to sample"

line 566, suggest replace "Especially" with "In particular"

line 582, suggest replace "Taken" with "Taking"

line 585, "organic matter" I think you want to be careful in your wording here, this reference concerns DOC export from the Arctic Ocean only; not DON, which may undergo quite different cycling pathways.

Figure S3, please note spelling in Fig "remineralization" versus the caption "remineralisation" and

"remineralization"

Figure S5, please note spelling in Fig "remineralisation" versus the caption "remineralization"

Reviewer #3 (Remarks to the Author):

This manuscript has been thoroughly and thoughtfully revised in response to reviewer comments. Uncertainties and caveats are now clearly addressed, allowing readers to focus more attention on the model results and less attention on the validity of various assumptions and methodological choices. My only suggestions at this point are:

- 1) Change the "Results" heading to "Results and Discussion", and remove the "Discussion" heading later in the manuscript.
- 2) On line 373, insert "combined" after "first".

Draft Only

Response to the reviewers

We thank the reviewers for their very helpful and constructive comments. In the following we address the reviewers comments point by point.

Reviewer 1

1.1 — Line 87: “river basin”

should be plural, right? river basins or watersheds

Reply: Changed as suggested.

1.2 — Check for consistent use of British or American English:

- remineralisation vs. remineralization
- remineralised vs. remineralized
- optimized...
- initialized...
- localised...
- hypothesised...

Reply: The manuscript was changed accordingly.

1.3 — I suggest to break this long sentence into two. Otherwise it’s hard to follow, especially with all the “thus” and “therefore”. For example, stop at Arctic Ocean.

Reply: The sentence was broken up in two sentences as suggested.

1.4 — This should be total organic carbon (TOC). The literature data that you use from Lantuit et al., 2012 is definitely TOC.

Reply: Changed as suggested.

1.5 — What about atmospheric deposition? You have mentioned it at another occasion.

Reply: Atmospheric deposition is not part of the external nutrient sources in the model and therefore not mentioned in the model description. The effect of not simulating atmospheric deposition is discussed in the main manuscript.

1.6 — Please rewrite to clarify that “NoTerr” neither includes riverine nor coastal fluxes. Right now it reads that it does not include riverine fluxes only...but that it might still include coastal fluxes.

Reply: Changed as suggested.

1.7 — Why “would contradict”? It does contradict. Then, in the coming sentences you address this contradiction. I suggest to remove “would”.

Reply: Changed as suggested.

1.8 — better cite like this: Brown, J., O. Ferrians, J. A. Heginbottom, and E. Melnikov. 2002. Circum-Arctic Map of Permafrost and Ground-Ice Conditions. Version 2. Boulder, Colorado USA: National Snow and Ice Data Center. [Date Accessed]. doi:10.3133/cp45

Reply: Changed as suggested.

Reviewer 2

2.1 — I still find the use of the term “data-based” quite confusing in the text, as it appears to be used interchangeably for in situ measurements and remotely sensed data, two types of data which I think should be distinguished when considering their use within (or in validation of) the model. I’d like to suggest that “data-based” be replaced with “remote sensing-based” where it is referring to satellite observations throughout the text; and replaced with “observation-based” (or another distinguishing term) where it is referring to extrapolations based on measured values. While I agree that these inputs are all observations and are all “data”, I think it is important that the reader be able to distinguish extrapolations based on in situ measured values (as your climatologies are) and those determined by proxy (satellite remote-sensing products). Please see my suggestions below (see minor suggestions) on specific instances where confusion arises.

line 84, as indicated in my previous review, the term “data-based” as used in the text is still very confusing. I suggest “data-based” be replaced with “remote sensing-based” where it is referring to satellite observations throughout the text; and replaced with “observation-based” (or another distinguishing term) where it is referring to extrapolations based on measured values. Specifically I would suggest:

line 84, replace “spatially resolved, data-based forcing set” with “spatially resolved forcing derived from in situ data, set”

in Table 2 title, at line 190, it is not clear here what “data-based” refers to, are the simulated values not determined based on the Arctic-GRO measured data? Perhaps just remove “and data-based”

in Table 2, final column of Table (right-hand column), this instance is referring to “remote sensing-based” correct?

in Figure 3, title to figure 3c, and in caption (line 204), replace with “remote sensing-based”;

in Figure 3, caption (line 202) I believe refers to the Arctic-GRO data, here I would suggest simplifying to “simulated Arctic Ocean NPP with nutrient input. . .” as it is clear your simulation is using the Arctic-GRO data

line 221, here I believe “observation based” is also referring to “remote sensing-based” estimates? (see why it’s very confusing?)

line 320, replace “data-based” with “remote sensing-based”

Figure 4, caption (line 329/330) replace “data-based” with “remote sensing-based”

line 366, replace “data-based” with “remote sensing-based”

line 403, here, again, “data-based” refers to measured values, here I would just remove “data-based” completely

line 608, here “Data-based” can be left as is since it’s referring to ALL the estimates based on data

line 610, here “observation based” is referring to “remote sensing-based” and should be changed

Supplement. Figure S5, line 43, replace “data-based” with “remote sensing-based”

Reply: Changes were made throughout the manuscript as suggested by the reviewer.

2.2 — I’m still concerned that the importance of coastal erosion is determined from a system where turbidity is not accounted for. To me this means that while you may have stimulated primary productivity by the nutrients released from coastal erosion in the model, in the real ocean the turbidity associated with coastal erosion may be too high to permit enough solar radiation for primary production to occur. This moves an

erosion dominated coastline from N-limitation to light limitation; which would not be accounted for in your model, producing an over-estimate of primary productivity. I think some more discussion on the implications of not accounting for turbidity is warranted.

Reply: As suggested by the reviewer, we have added the treatment of turbidity by the model to the brief model description in the Introduction and discussed possible effects in the Discussion :

“We also do not consider the addition of terrigenous particulate matter emanating from the rivers or coastal erosion and their effect on turbidity of the waters, and hence the light availability for phytoplankton growth” (Introduction)

“Furthermore, we consider our lack of consideration of the impact of the terrigenous particulate matter on turbidity to have only a minimal impact on our results and conclusions. First, turbidity from particles is globally⁴⁵ and in the Arctic Ocean^{46,47} mostly confined to the very nearshore zone and temporally to the spring breakup, as particulate matter from coastal erosion and rivers settles close to its origin^{34,48}. Given the large spatial extent of the Arctic shelves, particles thus only affect a minor part of these shelf seas. Second, while the particles settle quickly out of the euphotic zone, the nutrients remain, unleashing their impact downstream of the river mouth with some delay. Thus, while the consideration of the input of terrigenous particles would result in a spatial and temporal delay of the NPP’s response to an input of terrigenous nutrients, we do not expect any fundamental change in the magnitude of the response its magnitude.” (Discussion)

In addition, we have derived the photic depth in the Arctic based on MODIS-aqua remote-sensed attenuation coefficients that accounts for both chlorophyll and terrestrial particles and compared it to the simulated photic depth by NEMO-PISCES, which only takes into account chlorophyll and not particles(Fig. R1). In line with the previous studies now cited in the revised manuscript (Wegner et al., 2003; Shi and Wang, 2010; Klein et al., 2019), the comparison shows that the simulated photic depth is only overestimated in the very nearshore zone.

Figure R1: Photic zone (less than 1% of light left) in July simulated by NEMO-PISCES (right) and derived from remote-sensed attenuation coefficients from MODIS-aqua.

2.3 — The authors address my original question about the extrapolation of total annual fluxes to determine seasonal fluxes quite thoughtfully, I think it would benefit the paper to include some details of this explanation in the methods.

Reply: A more detailed explanation has been added to the Methods as suggested:

“By predicting the average annual fluxes (prediction of spatial variability only) first, these fluxes have lower uncertainty in the flux predictions than if spatial and seasonal variability had been predicted at the same time. Note that for the N-budget and NPP calculations for the Arctic Ocean, the mean average annual riverine input fluxes are of highest importance, and it is thus our priority to make the predicted mean annual fluxes as reliable as possible. The seasonal distribution of these fluxes over the year are of sub-ordinate importance.”

2.4 — line 49, suggest starting the sentence with “It has been estimated that. . .” as the studies reported are remote sensing-based observations and are therefore not accounting for changes throughout the water column

Reply: Changed as suggested.

2.5 — line 71, suggest replacing “exports” with “composition” as the exports are calculated whereas it is concentrations that are measured

Reply: “exports” were changed to “fluxes” consistent with the formulations throughout the manuscript.

2.6 — there have been time series observations carried out in several medium and small Arctic rivers, I think they should be mentioned here, including:

Li Yung Lung, J. Y. S., Tank, S. E., Spence, C., Yang, D., Bonsal, B., McClelland, J. W., & Holmes, R. M. (2018). Seasonal and geographic variation in dissolved carbon biogeochemistry of rivers draining to the Canadian Arctic Ocean and Hudson Bay. *Journal of Geophysical Research: Biogeosciences*, 123, 3371–3386. <https://doi.org/10.1029/2018JG004659>

McClelland, J. W., Townsend-Small, A., Holmes, R. M., Pan, F., Stieglitz, M., Khosh, M., & Peterson, B. J. (2014). River export of nutrients and organic matter from the North Slope of Alaska to the Beaufort Sea. *Water Resources Research*, 50, 1823–1839. <https://doi.org/10.1002/2013WR014722>

Holmes, R. M., Peterson, B. J., Gordeev, V. V., Zhulidov, A. V., Meybeck, M., Lammers, R. B., & Vörösmarty, C. J. (2000). Flux of nutrients from Russian rivers to the Arctic Ocean: Can we establish a baseline against which to judge future changes? *Water Resources Research*, 36(8), 2309. <http://doi.org/10.1029/2000WR900099>

Holmes, R. M., McClelland, J. W., Raymond, P. A., Frazer, B. B., Peterson, B. J., & Stieglitz, M. (2008). Lability of DOC transported by Alaskan rivers to the Arctic Ocean. *Geophysical Research Letters*, 35, L03402. <https://doi.org/10.1029/2007GL032837>

I suggest modifying the sentence beginning at line 71 to something along the lines of: “Measurements of riverine nutrient composition are taken frequently at the six largest Arctic rivers (Mackenzie, Yukon, Kolyma, Lena, Ob, Yenisei), and have been periodically recorded in several smaller river systems (e.g.,

Holmes et al., 2000, 2008; McClelland et al., 2014; Li Yung Lung et al., 2018), but few time series observations have been collected elsewhere.”

Reply: Changed as suggested.

2.7 — line 74, suggest modifying the sentence to read, “. . . fluxes from the six largest rivers were extrapolated. . . ”

Reply: Changed as suggested.

2.8 — line 75, suggest modifying the sentence to read, “Riverine nutrient inputs determined from previously published pan-Arctic organic carbon fluxes have been used to estimate riverine-driven NPP to vary between 4% and 10%...”

Reply: Changed as suggested.

2.9 — line 86, “(in terms of basin size)” this is confusing, in what ways are they not the six largest rivers?

Reply: This was deleted to avoid confusion.

2.10 — line 86, suggest including the caveat, “. . . Arctic rivers^{12,21}, five of which drain directly into the Arctic Ocean, which. . . ”

Reply: The suggested addition would have made the sentence too complicated. We have therefore decided not to make the suggested change. Instead we have added the following sentence to the Methods:

“Although the Yukon river does not drain directly into the Arctic Ocean, its large drainage basin lies predominantly north of 60°N and is representative of other ungauged Arctic rivers that drain directly into the Arctic Ocean.”

2.11 — line 87, suggest “using a spatially”

Reply: Changed as suggested.

2.12 — line 95, here it suggests you do use canonical Redfield ratios (106:16:1)? if not, you should state the ratios used here. I agree with the other reviewer that if you are not using Redfield ratios explicitly (C:N:P = 106:16:1) then you should not refer to them as such. You can simply use the term “stoichiometric ratios” or “C:N:P stoichiometry” in place of Redfield references.

Reply: Changed to “the stoichiometric marine ratios”.

2.13 — line 112, why is the Pechora River now part of the results? From the methods I understood that only data from the six largest Arctic rivers were included?

Reply: The Pechora is not used for the statistical extrapolation model. However, the extrapolated riverine input includes all Arctic Rivers and thus also the Pechora river. We mentioned it here as the extrapolation shows that it delivers an important amount of riverine nitrogen although it is not part of the six monitored rivers.

2.14 — Figure 2, please note spelling in Fig 2b “remineralization” versus the caption “remineralisation”

Reply: Changed to “remineralisation” throughout the manuscript.

2.15 — line 143, suggest “six largest Arctic rivers”

Reply: Changed as suggested.

2.16 — line 163, suggest “The estimate presented here of”

Reply: We preferred to keep the existing formulation.

2.17 — line 168, suggest removing the word “earlier”

Reply: Changed as suggested.

2.18 — line 196, why do the (No Coast) data not appear in Table 2? I think these should be included (even if they are just determined by a subtraction).

Reply: The data was added to Table 2 as suggested.

2.19 — line 224, suggest removing “Redfield” as “Assuming a C:N ratio of 122:16” is sufficient.

Reply: Changed as suggested.

2.20 — line 250, what are the pan-Arctic nutrient recycling rates you determine? it would be nice to have some ranges to compare to the literature values in the next sentence

Reply: The average rate is 7 and was added to the text.

2.21 — line 282, is 60% intended to be the average of 20-80%? (if so, it is incorrect)

Reply: 60% is the best estimate. The uncertainty towards lower remineralisation rates is higher, thus the uncertainty range is not symmetric.

2.22 — line 283, does “organic matter” refer to C or N or both?

Reply: It refers to nitrogen. The manuscript was changed accordingly.

2.23 — Figure 4, caption line 324, suggest remove “in dependence nitrogen” this looks like a typo

Reply: Changed as suggested.

2.24 — line 340, this is not actually changed as indicated in the response to reviewer file (see response 3.10), this sentence should begin, “The first study by Tank et al.,18. . .”

Reply: Now changed as suggested.

2.25 — line 344/345, again, I'm concerned that these coastal erosion numbers derive from a system where turbidity is not accounted for (see notes above). I think this should be included as a possible area of uncertainty.

Reply: The uncertainty was added in the Methods. As the Arctic Ocean remains nitrogen limited (Supplementary Figure 1), a difference in light availability would likely not change the results significantly.

2.26 — line 374, suggest “coastal erosion on Arctic Ocean”

Reply: Changed as suggested.

2.27 — line 377, it might be worth including a note that coastal erosion is limited in extent and has its main impact immediately adjacent to the coast

Reply: We do not agree here. Although the immediate impact of coastal erosion is adjacent to the coast, the 7-fold remineralisation transports nitrogen molecules from coastal erosion far away from the coast. Thus nitrogens from coastal erosion can have an impact over the entire shelf regions.

2.28 — line 404, suggest “. . . estimates determined over the first decade of the 21st century from the six largest Arctic rivers. . .”

Reply: Changed as suggested.

2.29 — line 407, I think a caveat about the Yukon River is warranted here, as I do think it should be included in your data set (it's important for scaling), but as the other reviewer noted it does not drain into the AO directly. I suggest adding something like the following at line 407, “We note that although the Yukon River does not drain directly into the Arctic Ocean, its large drainage basin lies predominantly north of 60N and is representative of other un-gauged Arctic regions that drain directly into the Arctic Ocean. . .”

Reply: Changed as suggested.

2.30 — line 416, suggest “six largest Arctic rivers”

Reply: Changed as suggested.

2.31 — line 422, I think more explanation is needed here as to why a global runoff data set was used to extrapolate C_T and A_C fluxes, especially as this was not done for other nutrients

Reply: For C_T and A_T , statistical models for the extrapolation of observed riverine fluxes to the pan-Arctic were previously developed by Tank et al using global runoff data. We decided to use the same statistical models as Tank et al. and geodata sets of predictors. The obtained fluxes were then divided by the runoff data (Fekete et al., 2002) to obtain C_T and A_T concentrations for every Arctic river. These were then multiplied by the runoff data used in the ocean-biogeochemical model NEMO-PISCES (Dai and Trenberth, 2002)

. For DOC and dissolved nutrients, for which no statistical models existed beforehand, multiple linear regression was used to predict concentrations directly. As for C_T and A_T , these concentrations were multiplied by the runoff data used in the ocean-biogeochemical model NEMO-PISCES to obtain the respective fluxes.

We have clarified this in the revised manuscript in the Methods.

2.32 — line 451, suggest replace “in consistency” with “consistent” and line 458, suggest replace “in consistency” with “consistent”

Reply: Changed as suggested.

2.33 — line 465, suggest “six largest Arctic rivers”, I’ve made this suggestion a few places as reading “six big rivers” does not indicate that these rivers are the largest, you could use “big-6” as is done in Holmes et al.’s work, but I think saying “six largest Arctic rivers” is more appropriate

Reply: Changed to “six largest Arctic rivers” as suggested.

2.34 — line 467, I hadn’t thought of this until noticing the inclusion of the Pechora River earlier, but how do you determine WHERE along the Arctic coast these inputs occur? (and does it matter?) For example, are the inputs from all ungauged areas in NA distributed evenly across the coast or is there one point source (like the Mackenzie) where these materials are brought into the model ocean? Perhaps just adding a line or two about how NEMO-PISCES treats these additions.

Reply: The nutrient fluxes and coastal erosion fluxes were added at their geographical location. In brief we have extrapolated riverine and coastal erosion on a 2-D field ($1^\circ \times 1^\circ$). Each river or coastal segment was assigned to its nearest grid cell on this field. This monthly 2D-field was then used to force the ocean. A detailed description is provided in the Methods section “Ocean-biogeochemical model NEMO-PISCES”:

“Nutrient fluxes from rivers and coastal erosion were calculated from the newly generated monthly 2-D forcing files. The fluxes from the regular 1° grid were rescaled to the curvilinear model grid. To do so, each model grid cell was assigned to its nearest grid cell on the 1° grid. The flux from the 1° grid was then divided between the assigned model grid cells, for coastal erosion in proportion to the surface area of each cell and for rivers in proportion to the runoff in each cell. This approach ensures the conservation of mass of the calculated fluxes and correlates runoff with carbon and nutrient fluxes. Vertically, the river input was divided on the first two vertical model grid levels (0–13 m) to account for a finite river depth. ”

2.35 — line 498, here you include a note about possible shading effects of particles, as above, I think some more thought to the implications is warranted in the discussion

Reply: See response above.

2.36 — line 498, suggest remove “Redfield” and line 523, suggest remove “Redfield”

Reply: Changed as suggested.

2.37 — line 522, is this the marine stoichiometric ratio you refer to below? suggest “a marine stoichiometric C:N:P ratio”

Reply: Changed as suggested.

2.38 — line 526, does PISCES use the canonical Redfield ratio (106:16:1) or the marine ratio listed above (122:16:1) from Takahashi? If it uses Redfield then you can leave this in, however if not you could replace “Redfield” with “stoichiometric”

Reply: Changed as suggested.

2.39 — line 550, suggest replace “hostile” with “difficult to sample”

Reply: Changed as suggested.

2.40 — line 566, suggest replace “Especially” with “In particular”

Reply: Changed as suggested.

2.41 — line 582, suggest replace “Taken” with “Taking”

Reply: Changed as suggested.

2.42 — line 585, “organic matter” I think you want to be careful in your wording here, this reference concerns DOC export from the Arctic Ocean only; not DON, which may undergo quite different cycling pathways.

Reply: Changed as suggested.

2.43 — Figure S3, please note spelling in Fig “remineralization” versus the caption “remineralisation” and “remineralization” and Figure S5, please note spelling in Fig “remineralisation” versus the caption “remineralization”

Reply: Changed as suggested.

Reviewer 3

3.1 — Change the “Results” heading to “Results and Discussion”, and remove the “Discussion” heading later in the manuscript.

Reply: Changed as suggested.

3.2 — On line 373, insert “combined” after “first”.

Reply: Changed as suggested.

References

- Dai, A., & Trenberth, K. E. Estimates of freshwater discharge from continents: Latitudinal and seasonal variations. *Journal of hydrometeorology* **3.6**, 660–687 (2002).
- Fekete, B. M. et al. High-resolution fields of global runoff combining observed river discharge and simulated water balances. *Glob. Biogeochem. Cycles* **16**, 1–10 (2002).
- Klein, K.P. et al. Long-Term High-Resolution Sediment and Sea Surface Temperature Spatial Patterns in Arctic Nearshore Waters Retrieved Using 30-Year Landsat Archive Imagery. *Remote Sens.* (2019).
- Shi, W., and Wang, M. Characterization of global ocean turbidity from Moderate Resolution Imaging Spectroradiometer ocean color observations, *J. Geophys. Res.*, **115**, C11022 (2010).
- Wegner, C., et al. Suspended particulate matter on the Laptev Sea shelf (Siberian Arctic) during ice-free conditions. *Estuar. Coast. Shelf S.* **57.1-2**, 55-64 (2003).